# Oncogenic Signaling in Tumorigenesis and Applications of siRNA Nanotherapeutics in Breast Cancer

**DOI:** 10.3390/cancers11050632

**Published:** 2019-05-06

**Authors:** Nur Izyani Kamaruzman, Noraini Abd Aziz, Chit Laa Poh, Ezharul Hoque Chowdhury

**Affiliations:** 1Department of Biomedical Science, Faculty of Medicine, University of Malaya, 50603 Kuala Lumpur, Malaysia; nurizyanikamaruzman@gmail.com; 2Centre for Virus and Vaccine Research (CVVR), Sunway University, 47500 Subang Jaya, Selangor, Malaysia; norainiaa@sunway.edu.my (N.A.A.); pohcl@sunway.edu.my (C.L.P.); 3Jeffrey Cheah School of Medicine and Health Sciences, Monash University Malaysia, 47500 Subang Jaya, Selangor, Malaysia

**Keywords:** breast cancer, siRNA, cell signaling, active targeting, passive targeting, EPR effect, oncogenes, nanoparticles, nanomedicine

## Abstract

Overexpression of oncogenes and cross-talks of the oncoproteins-regulated signaling cascades with other intracellular pathways in breast cancer could lead to massive abnormal signaling with the consequence of tumorigenesis. The ability to identify the genes having vital roles in cancer development would give a promising therapeutics strategy in combating the disease. Genetic manipulations through siRNAs targeting the complementary sequence of the oncogenic mRNA in breast cancer is one of the promising approaches that can be harnessed to develop more efficient treatments for breast cancer. In this review, we highlighted the effects of major signaling pathways stimulated by oncogene products on breast tumorigenesis and discussed the potential therapeutic strategies for targeted delivery of siRNAs with nanoparticles in suppressing the stimulated signaling pathways.

## 1. Introduction

Breast cancer is one of most common life-threatening cancers and the second leading cause of female deaths worldwide. About 1.67 million new cases of breast cancer were diagnosed in 2012 worldwide [1]. According to the data revealed by the American Cancer Society, around 266,120 new cases of invasive breast cancer will be diagnosed in American women in 2018 [2]. The statistics emphasized that one in every eight women in the United State of America is at risk of having breast cancer. Based on the Malaysian National Cancer Registry Report (2007–2011), 1 in 30 females is at risk of having breast cancer in a lifetime [3]. The mortality rate of breast cancer in Malaysia is estimated to be ~16.7 to 20 in 100,000 [4].

Breast cancer is the malignant cell growth that originates from the breast cells at the inner lining of the breast ducts or lobules that supply milk [5]. There are stages of breast cancer diagnosis, where at stage 0, the cancer cells are found to be localized at the lobules or ducts of the breast. At stage I, II, and III, the cancer may be defined by the size of the tumors and the area that the cancer cells have spread, such as the chest wall, skin, or the lymph nodes surrounding the breast. At the advanced or metastatic stage (stage IV), the cancer cells have metastasized to other organs or lymph nodes that are further away from the breast [5,6]. Breast cancer is a heterogeneous disease, as there are many distinct genes being overexpressed and acting as key players in the progression of the breast cancer cells [7]. The expression of breast cancer markers, such as estrogen receptor (ER), human epidermal growth factor receptor 2 (HER2/neu), progesterone receptor (PR), and urokinase plasminogen activator (uPA) has been used to evaluate the progression and aggressiveness of the disease [7,8,9]. The untreated lesion of the ducts or lobules may lead to proliferation and formation of metastatic cells that can develop the ability to invade blood and lymphatic vessels and metastasize to other parts of the body, such as brain, lung, liver, and bones [10]. The common symptom of the disease is the formation of lumps in the breast. Other than that, patients may also experience changes in the breast’s features such as thickening, swelling, distortion, tenderness, skin irritation, redness, nipple abnormalities, and discharge. 

Factors that have been associated with increased risk of breast tumorigenesis are sex, age, family history, breast condition, and endogenous estrogens. Females are more frequently diagnosed with breast cancer than males. The risk also increases with age, and postmenopausal women have been considered to have more risk. Women and men with first-degree relatives with breast cancer are shown to be at higher risk of getting the disease compared to those without family history of breast cancer. The mutations of the well-known tumor suppressor genes, *BRCA1* and *BRCA2*, are frequently associated with breast cancer. The faulty genes impairing the DNA repair process increase the chances of breast cancer. Conversion of proto-oncogenes into oncogenes via mutations is one of the prominent causes of the disease, promoting overexpression of growth factor receptors and subsequent cross-talks among their downstream signaling cascades, and can lead to proliferation and survival of cancer cells [11]. Besides that, an increase in mammographic breast density indicates a higher chance of the individual to develop breast cancer [12]. The presence of fat tissues, which can be the source of cholesterol, may increase the production of estrogens in high-density breast. Aromatase is the enzyme that promotes the production of estrogen from the androgens [13]. Other than that, there are studies showing that higher levels of estrogens are associated with the development and progression of breast cancer [14].

Biopsy taken from the mass formed in the breast confirms the presence of malignancy through laboratory screening. To date, there are few treatments of breast cancer such as surgery, radiotherapy, chemotherapy, hormonal therapy, and monoclonal antibody therapy [6]. The surgical procedure involves the removal of the tumors localized in the tissue, and mastectomy, which is the removal of the affected breast. Radiotherapy uses high-energy rays that kill the actively dividing cancerous cells [15]. Chemotherapy for breast cancer is a treatment by delivering cytotoxic drugs either through intravenous injection or oral delivery, allowing the drugs to travel through the blood circulation before reaching the cancer cells [6]. Monoclonal antibody therapy for breast cancer attempts to trigger the immune system to destroy the cancer cells by allowing binding of the antibody with the antigens that are present on the cancer cells. For example, introduction of traztuzumab that targets HER2 on breast cancer cells causes arrest of cancer cells at the G1 phase of the cell cycle, thus reducing the process of cell proliferation. Further, it may down-regulate the expression of HER2 and reduce the dimerization of the receptor [16,17]. Hormonal therapy is performed, for instance, by using estrogen antagonist that blocks the action of estrogen, such as tamoxifen and raloxifene or aromatase inhibitors (anatrozole and exemastene) [6]. As research showed that 70% of the breast cancer cases are estrogen-receptor positive, the usage of estrogen antagonist and aromatase inhibitor is commonly employed to treat breast cancer. As breast cancer is a heterogenous disease, there is rapid growth of ongoing research on developing breast cancer therapeutic strategies to encounter the likely cause of the disease. Triple-negative breast cancer (TNBC) defines the disease without or having less expression of the well-known breast cancer markers like ER, PR, and HER/neu [18], therefore, requiring different treatment approaches. Glycotherapy is one of the potential strategies to target aberrant glycosylation that promotes abnormal cellular activities and carcinogenesis [19]. The aspragine-linked (N-linked) glycoprotein was found to have roles in the progression of breast cancer, such as angiogenesis [20]. Banerjee et al. (2011) has shown that Tunicamycin is able to inhibit angiogenesis in vitro and in vivo. They have observed reduced expression of vascular endothelial growth factor receptors (VGFRs) and N-glycan in the tumor micro-vessels [21]. Another field of research is immunotherapy, which comprises, for instance, targeting the pathway of programmed death 1/ programmed death ligand 1 (PD-1/PD-L1), which involves responses from T-cells [22]. PD-L1 that is expressed by the tumor cells binds to PD-1 proteins expressed by T-cells. The interaction of PD/PD-L1 inhibits T-cells from killing the tumor cells. Atezolizumab (Tecentriq ^TM^) is known as a checkpoint inhibitor and functions as an anti-PD-L1 monoclonal antibody. The inhibition of PD-1/PD-L1 binding enables the killing of tumor cells by T-cells. There are also studies on complementary and alternative medicines (CAM) as a treatment option of breast cancer, such as Ayuverda (traditional Indian medicine) and traditional Chinese medicine [23]. Several patients opted for CAM as they have experienced failures in other treatments. CAM typically employs herbal and botanical therapy (e.g., homeopathy), mental therapy (e.g., meditation and hypnosis), and physical therapy (e.g., acupuncture, massage, yoga, and Chi Gong). Thus far, there lacks scientific evidence of CAM, with no successful clinical trials reported in effectively curing the disease [24].

Numerous short- and long-term effects from the chemotherapy such as risks of cardiac toxicity, development of secondary cancer, neurotoxicity, premature menopause, and effects on sexuality with high costs of drugs and treatments have led to emotional, physical, and financial burden for the patients and the community [25]. Thus, it is crucial to identify key players of the disease’s development and to develop effective therapeutic strategies in targeting breast cancer with or without minimal side effects.

## 2. Signaling Pathways and Oncogene Involvement in Breast Cancer

Cell membrane receptors and ion channels receive stimuli such as hormones, neurotransmitters, antibodies, cytokines, growth factors, and ions from the extracellular region that influence cell signaling [26]. The interactions between the stimuli and the receptors or ion channels may trigger various downstream signaling pathways, such as Mitogen-activated protein kinase (MAPK) and phosphoinositide-3-kinase–protein kinase B (PI3K/AKT) and Ca^2+^ signaling pathways at the intracellular level (Figure 1). The interplay or cross-talks between the signal transduction pathways build up the complexity of breast cancer signaling cascades, thus, complicating the process of curing the disease. The interconnected signaling pathways may induce the breast cancer cells to proliferate and survive under a heterogeneous condition with various up-regulated and down-regulated proteins. Proto-oncogenes and tumor suppressor genes are involved in the maintenance of the normal cell functions such as growth, division, and survival. However, mutations of these genes in the form of deletions, insertions, or substitutions, resulting in gain or loss of functions, may constitutively activate the signaling pathways, initiating the tumorigenesis [27,28]. Thus, identifying the oncogenes and tumor suppressor genes governing the breast cancer signaling pathways is an important goal in therapeutic intervention of breast cancer.

### 2.1. Mitogen-Activated Protein Kinase (MAPK) Pathway

Gene therapy has been advocated to treat cancers. There are many genes and proteins that are being up- or down-regulated in the signaling pathways, thus promoting the proliferation and survival of breast cancer cells. For example, mitogen-activated protein kinases (MAPKs) are the proteins that function in delivering and amplifying the extracellular signals. Researchers have identified six different groups of MAPKs, which are extracellular signal-regulated kinases (ERK)1/2, ERK3/4, ERK5, ERK7/8, Jun N-terminal kinase (JNK)1/2/3, and the p38 isoforms α/β/γ (ERK6)/δ [29,30,31]. MAPK signaling through the ERK pathway activation is regularly being activated via the binding of ligand with the cell membrane receptor, such as the receptor tyrosine kinase (RTK). These will later promote the downstream responses of the pathway, for example, the activation of Ras protein. The activation of Ras may lead to subsequent stimulation of ERK1/2 signaling proteins to transmit signals into the nucleus for gene transcription and expression; thus, cell proliferation, survival, apoptosis, and differentiation are turned on [32]. Hyper-expression of MAPK was found in the primary human breast cancer tissue compared to the benign portion, which is related to metastatic potential of the disease [33]. Studies on down-regulating the expression of MAPK have shown the decrease in breast cancer cell proliferation and migration [34].

### 2.2. PI3K/AKT Pathway

AKT or protein kinase B is also an important factor in regulating the cell proliferation, survival, glycogen metabolism, and motility [35]. It exists in three isoforms in mammals; AKT1, AKT2, and AKT3. Phosphatidylinositol 3-kinase (PI3K) is the vital protein in connecting the signals from the cognate receptor tyrosine kinase towards AKT [36]. Numerous studies have revealed the deregulation and mutations of genes of this pathway in 70% of the breast cancers [37,38]. Upon stimulation by ligand binding to the receptor tyrosine kinase, PI3K signaling pathway may be activated, thus transferring the message down to the AKT pathway, such as the mTOR signaling pathway. The phosphorylation of AKT (pAKT) promotes cellular functions, such as proliferation and survival. However, overexpression of pAKT proteins was found in 33% of ductal carcinoma in situ and in 38% of the invasive breast cancer cases through the immunohistochemistry of tissue microarray [39]. In another study involving siRNA knockdown targeting the AKT in MCF-7 human breast cancer cell line, the introduction of the siRNA has reduced the expression of AKT and BCL-2 (anti-apoptotic protein) proteins, which may enhance the probability of cancer cell death [40].

### 2.3. Calcium Signaling Pathway

Calcium ion (Ca^2+^) is known to be a ubiquitous cellular signal and is one of the important second messengers in cell signaling. It is crucial to maintain its homeostasis in normal cell signaling series. Ca^2+^ is released in cytosol either from internal stores, such as endoplasmic reticulum or from external medium through different cell membrane-associated channels, through the action of Ca^2+^ itself, intracellular messengers, such as inositol-1,4,5-trisphosphate, or the status of intracellular Ca^2+^ storage. It may initiate different types of protein activation or phosphorylation and changes in protein shapes and charges, which may subsequently vary the interactions with other respective components [41]. Moreover, Ca^2+^ plays a role in cell proliferation, as it is involved with the activation of the cyclin-dependent kinases (CDK4 and CDK2) for the progression of the cell cycle from the G1 to S phase [42]. In the pathological environment, malignant cells may acquire the six hallmarks of cancers, as described by Hanahan and Weinberg (2000): (a) Self-sufficiency in growth signals, (b) insensitivity to growth-inhibitory signals, (c) evasion of programmed cell-death (apoptosis), (d) limitless replication potential, (e) sustained angiogenesis, and (f) tissue invasion and metastasis [43]. Any disturbance of Ca^2+^ homeostasis may alter the cell cycle progression and trigger the emergence of one or more of the cancer hallmarks. Di et al. (2015) [44] showed that the overexpression of Rap2B, a GTP-binding protein, increased the intracellular calcium level, thus later promoting the phosphorylation of ERK1/2 in Bcap-37 and MDA-MB-231 breast cancer cells. They also observed increase in proliferation, migration, and invasion of the cancer cells [44].

Endoplasmic reticulum release of Ca^2+^ and subsequent uptake by mitochondria involves programmed cell death. Nevertheless, increases in Ca^2+^ influx activates the survival signaling pathways of the cancer cells. Cancer cells develop an antioxidant system against the reactive oxygen species (ROS) such as hydrogen peroxide (H_2_O_2_) to maintain cells’ activities. H_2_O_2_ produced by mitochondria mediates cysteine oxidation on transient receptor potential ankyrin 1 (TRPA1). TRPA1, a cation channel on the cell membrane, enables the up-regulation of Ca^2+^ into the cellular region and activates the anti-apoptotic pathway such as the PI3K/AKT signaling pathway [45]. Data analysis from the cancer genome atlas found the overexpression of TRPA1 in breast cancer.

### 2.4. Notch Signaling Pathway

Notch signaling pathway begins with the interaction of the DSL (Delta/Serrate/LAG-2) ligands on one cell and Notch receptor on the adjacent cell [46]. The signaling pathway is associated with the cellular progression, such as proliferation, apoptosis, angiogenesis, hypoxia, cancer stem cell activity, epithelial to mesenchymal transition (EMT), and metastasis. In breast cancer, Notch receptors and their ligands were found to be overexpressed. Notch receptors have been categorized into four groups, Notch1 to Notch4; while the DSL ligands, which are transmembrane ligands, have five groups (Jagged1, Jagged2, Delta-like1, Delta-like3, and Delta-like4). The expressions of *cyclinA*, *cyclinB*, and *cyclinD1* genes were found to be upregulated in Notch signaling cascade, while the survival of breast cancer cells might be induced via AKT pathway activation by Notch signaling pathway [46,47,48].

### 2.5. Hedgehog Signaling Pathway

Hedgehog signaling pathway controls the process of cell proliferation, survival, differentiation, tissue homeostasis, regeneration, and stem cell maintenance [49]. Most of the basal-like breast cancers (BLBC) have the triple negative phenotype of the important receptors (ER−, PR−, and HER2) and are resistant to chemotherapy treatments. The BLBC has an aggressive growth and has the possibility to metastasize to other organs. Mott et al. (2018) has shown that the forkhead-box transcription factor C1 (FOXC1) plays a role in 4T1 murine metastatic breast cancer cell proliferation, migration, and invasion, although no significant effects were reported in the in vivo study [50]. FOXC1 was found to be overexpressed in BLBC and was able to activate the hedgehog signaling pathway [51]. In another case, this pathway might be initiated by the interaction of modified Hedgehog ligand towards the patched (Ptch1) receptor, a 12-pass transmembrane receptor. This event led to the activation of smoothened (Smo), a seven transmembrane protein that further stimulated multi-complex proteins that contained Gli protein. A zinc finger transcription factor then traveled into the nucleus to initiate the transcription of targeted genes [52,53]. Overexpression of Gli 1 protein (belonging to the family of Gli transcription factors) was observed to be associated with the unfavorable prognosis and survival of the breast cancer cells [54,55].

### 2.6. JAK/STAT Signaling Pathway

Extracellular stimuli such as cytokines (e.g., interleukins, interferons, and growth factors) can activate the JAK/STAT signaling pathway. JAK or Janus kinase and STAT (signal transducers and activator of transcription) are the intracellular proteins that cooperate with the transmembrane receptor in conveying signals down to the nucleus for DNA transcription and gene expression. STAT acts as the substrate of JAK, becomes phosphorylated, and travels into the nucleus to promote gene transcription. JAK/STAT signaling pathway is involved in stem cell maintenance, hematopoiesis, and participate in the process of inflammatory response. This signaling pathway may promote cell proliferation, differentiation, and has a role in controlling cellular apoptosis. The suppressor of cytokine signaling proteins (SOCS) is the regulator of the JAK/STAT negative feedback loop that functions as competitive inhibitors to STAT while STAT is the stimulator of the transcription of *SOCS* genes [56]. Dolled-Filhart et al. (2003) showed that STAT3 (a protein under the STAT family) and phosphorylated-STAT3 were overexpressed in 69.2% of breast cancer tumors [57]. Other investigations of STAT3 have confirmed the involvement of this protein in breast cancer malignancies [58,59].

### 2.7. Anti-Apoptotic Signaling Pathway

The anti-apoptotic signaling pathway is another pivotal component in breast cancer maintenance. BCL-2, BCL-XL, BCL-W, MCL-1, and BFL-1/A1 are the anti-apoptotic proteins in the BCL2 family. These gene are overexpressed in many cancers such as prostate, lung, stomach, ovarian, and breast carcinoma [60]. The BCL-2 (B-cell lymphoma 2) protein plays an anti-apoptotic role, leading to prolonged cancer cell survival [61,62,63,64]. Activation of the growth factor receptors, such as HER2, could modulate expression of BCL-2 via activation of PI-3 kinase signaling [63]. The cross-talks between the estrogen receptor (ER) with other membrane receptors might induce the transcription of target genes such as *BCL-2* gene expression. BCL-2 functions by inhibiting the pro-apoptotic proteins (e.g., BAD and BAX) in inducing cell death, thus, prolonging the survival of the cancer cells [64]. 

## 3. Growth Factor Receptors and Breast Cancer

The expression of the growth factor receptors (GFRs) is an important regulatory element that contributes to cell proliferation and survival. Regularly, the GFRs require ligand binding in order to transmit the downstream commands. The ligands may exist in the forms of growth factors, cytokines, or hormones. 

### 3.1. Epidermal Growth Factor Receptor (EGFR)

The EGFR is a transmembrane receptor that falls under the receptor tyrosine kinase family [10]. The EGFR family consists of four sub-proteins which are EGFR1 to 4 (also known as ErbB 1 to 4). EGFR is regularly activated by the EGF, which acts as the ligand and the downstream signaling cascades such as the Ras/Raf, MAPK, and the PI3K/AKT which may be stimulated to govern cell proliferation and survival [65]. EGFR is one of the receptors that has been associated with the progression of breast cancer and the overexpression of the EGFR is often associated with poor prognosis [66]. Price et al. (1999) has worked on the MDA-MB-231 breast cancer cell line and found that the EGF might stimulate the migration of the breast cancer cells via the activation of ERK1/2 signaling pathway [67,68]. In the case of molecular apocrine breast cancer (MABC), this molecular subtype of breast cancer is associated with poor prognosis as this subtype has negative expression of the estrogen receptor. Liu et al. (2018) [69] has found 53% of the MABC and non-MABC cases to be positive with EGFR expression via immunohistochemical analysis. The expression of EGFR was alongside the expression of other prominent breast cancer biomarkers such as the androgen receptor and Ki67 protein (the cellular marker for cell proliferation), thus suggesting EGFR is another therapeutic target for breast cancer treatment [69].

### 3.2. Insulin-Like Growth Factor 1 Receptor (IGF1R)

The IGF1R is a receptor that belongs to the IGFR family. It is a heterodimeric cell membrane receptor that comprises α- (subunit binding site) and β-(linked to the tyrosine kinase domain) chains that are projected towards the extracellular compound; whereas the tyrosine kinase domains are embedded within the layers of cell membrane [11,70]. The binding of IGF1 to the IGF1R results in the auto-phosphorylation of the tyrosine kinases and further activates the downstream signaling cascades, such as the PI3K/AKT and MAPK pathways [65]. The IGF family members, together with the IGF1R, were found to be overexpressed in breast cancer tumors and associated with cancer progression [71].

### 3.3. Transforming Growth Factor-Beta Receptor (TGF-βR)

TGF-βR has the TGFβ as the protein ligand which is available in the extracellular matrix (ECM) in order to encourage the intracellular signaling pathways [11]. There are three types of TGF-βR which are TGF-βR1, TGF-βR2, and TGF-βR3. The TGFβ1 and TGFβ3 ligands, once activated, will bind to the TGF-βR2, while the TGFβ2 has more affinity towards the TGF-βR3. The interaction between TGF-βR2 and its ligand may encourage the activation of TGF-βR1 [72,73]. Busch et al. (2015) showed that loss of expression of TGF- βR2 in the mammary fibroblast might stimulate tumorigenesis with increased tumor volume in the mouse xenograft model [74].

### 3.4. Vascular Endothelial Growth Factor Receptor (VEGFR)

The VEGFR is also a tyrosine kinase receptor that has seven immunoglobulin (Ig)-like domains projected at the extracellular region of the cell. The tyrosine kinase domains are rooted within the cell membrane layers (11). The VEGF–VEGFR interaction has often been associated with the angiogenesis or vasculogenesis of blood vessel in tumors, where the enlargement or growing tumors are in need of more nutrients supply as rapid growth rate is usually observed [75]. It has been shown that the growth of breast tumor in the murine model benefited from the expression of VEGFR1 expression [76]. VEGFR2 was confirmed to have roles in breast cancer angiogenesis and the inhibitor [YLL545, a novel synthesized compound from commercially available 1H-pyrazolo[3,4-d]-pyrimidin-4(5H)-one [1] for VEGFR2 has been shown to inhibit the downstream signaling regulators such as phosphor STAT and phosphor ERK1/2 [77]. Another compound named isomangeferin (a xanthone C-glucoside), was shown to bind to the VEGFR2 and suppressed tumor growth, metastasis, and angiogenesis [78].

### 3.5. Human Epidermal Growth Factor Receptor 2 (HER2/ERBB2)

HER2 is one of the members of receptor tyrosine kinase family that is encoded by *ERBB2* gene. The transmembrane receptor plays important roles in various cellular functions such as cell growth and differentiation [79,80]. However, 20%–30% of amplification of this gene is often seen in tumors of breast cancer [81]. Cross-talks between HER2 and other cell membrane receptors such as estrogen receptor (ER), IGFR, and EGFR may initiate downstream signaling through MAPK and PI-3 kinase pathways in breast cancer cells [82], suggesting the importance to determine the genes crucial in the cross-talks for therapeutic purpose. Moreover, there are other molecules that assist HER2 in governing breast cancer progression. The epithelial cell adhesion molecule (EpCAM) is a membrane glycoprotein that functions as a cell adhesion molecule in normal cells. However, it has been observed to be highly expressed in cancers including breast cancer [83,84,85]. N-glycosylation of the EpCAM was observed in parallel with overexpression of HER2 in breast cancer tissues. Furthermore, this event was shown to increase cancer cell proliferation and prevent apoptosis [86]. A study by Peiris et al. (2017) has shown that the co-translational modification of the glycans, such as the N-linked glycans, reduced the binding efficiency of Herceptin towards HER2, thus decreasing the competency of the treatment [87]. Therefore, HER2 and associated glycoproteins could be targeted with an aim to reduce breast cancer progression. 

## 4. siRNA Silencing Technique

Malignancies are often being associated with up-regulation of genes that causes overexpression of oncogenes [88,89,90,91]. Genetic manipulation, such as the introduction of the small interfering RNAs (siRNAs) has become one of the promising therapeutic approaches that is rapidly expanding. siRNA is a duplex RNA of 21–28 nucleotides that selectively degrades a mRNA transcript and thereby blocks its translation into a particular protein [92]. In eukaryotes, protein-coding genes are transcribed by RNA polymerase II to produce pre-mRNA which, upon further processing, becomes the mature mRNA [93]. The mature mRNAs travel from the nucleus into the cytoplasm for protein translation by ribosomes. The introduction of exogenous siRNA into the cells results in the formation of the RNA-induced silencing complex (RISC) by assembling with other proteins such as Argonaute and Dicer (Figure 2). Argonaute proteins are then activated to cleave the siRNA to become single stranded. In the cytoplasm, RISC carrying the single stranded siRNA binds to the complementary sequence on the targeted mRNA in a sequence specific manner. Slicer or Argonaute proteins will then cleave the mRNA complementary to the antisense strand in the newly formed double-stranded RISC-mRNA complex. The cleaved mRNA strands are recognized by the cell as aberrant and destroyed; thus, the expression of the targeted gene has successfully been ‘silenced’ [94,95].

### 4.1. Advantages of siRNA Delivery

As a promising cancer therapeutic strategy, siRNA has several potential advantages over chemotherapeutic drugs. Firstly, siRNA has a high degree of safety, since it inhibits the post-transcriptional stage of gene expression through complementary base pairing with a target mRNA without interacting with chromosomal DNA; thereby, the risks of mutation and teratogenicity are less [96]. The second advantage is its high degree of specificity in targeting a particular mRNA through RNA interference system, with the unlimited choice of targets. Another crucial advantage of siRNA is its high efficacy, suppressing the expression of a target gene strikingly in a single cancer cell with just several copies. Synthesis of siRNA is also much cheaper than that of antibodies or proteins.

### 4.2. Limitations of siRNA Delivery

There are several limitations to the clinical applications of siRNAs as therapeutics. The main challenge is the difficulty in passively delivering the siRNA which carries a negative charge due to the strong anionic phosphate backbone. The cell membrane which carries net negative charges repels the anionic siRNA, thus causing the process of passive diffusion of the exogenous siRNA to be challenging [97]. The water-soluble characteristic of the siRNA has further added hurdles to the process [98]. Despite the remarkable potency of siRNA in silencing specific gene expression, its half-life is too short because of the risk of degradation by serum nucleases which could affect the stability of siRNA, and quick elimination of the degraded products through the kidneys. In addition, ‘naked’ siRNA is hardly able to penetrate the tissue owing to its negatively charged phosphate backbone that could be repelled by anionic extracellular matrix molecules [99]. Even though siRNA molecules enter the cell through endocytosis, the fusion of endosomes with lysosomes results in degradation of the entrapped siRNAs. Nonetheless, various nanotechnology approaches have been harnessed to design suitable carriers for the siRNA to avoid degradation and assist in cellular delivery. Therefore, a suitable nano-carrier to transport siRNA molecules into tumor cells via endocytosis and subsequently release them in the cytoplasm is the prerequisite for achieving the maximum therapeutic outcomes from siRNA-mediated cleavage of the targeted mRNA. However, nanoparticles are prone to interact with reticuloendothelial system (RES) and hence, require surface modification prior to being used for systemic delivery of siRNA.

## 5. Delivery Systems of Potential Therapeutic siRNAs

Nanoparticles have emerged in the last few years as an alternative material for advanced diagnostic and therapeutic applications in medicine. A nanoparticle-based drug delivery system has two main targeting systems: Passive and active. Passive targeting relies on enhanced permeability and retention (EPR) effects of leaky vasculature (Figure 3) [99]. Tumor formation leads to underdevelopment of blood vessels that impairs the lymphatic drainage especially surrounding the tumor site. The leakiness of neoplastic blood vessels has great benefit in delivering and accumulating drugs up to 400 nm of size onto target sites [100]. Hobbs et al. (1998) showed that the vasculature of the tumors that was induced subcutaneously in mice had cut-off pore sizes in the range of 200 nm to 1.2 µm [101]. Active targeting includes ligand-mediated targeting, where ligands such as peptides or antibodies with affinity towards the nanoparticles or/and drugs are incorporated. The complex may recognize the targeted cells through binding to the receptors present on the cell surface. Active targeting may further augment the drug-delivery process to be more specific [100,102]. Therefore, it is crucial to identify highly expressed receptors, particularly on the breast cancer cells, to increase their specific binding with the ligands on the siRNA complexes and successful delivery of siRNA to the targeted sites.

Nanoparticles with size ranging 1–100 nm in diameter and large surface area [103] have been designed to bind and deliver siRNA, since naked siRNA is prone to degradation by serum nucleases and clearance by kidneys. Hence, the siRNA needs to be encapsulated with nanoparticles. Moreover, as mentioned above, nanoparticles can also be employed for targeted delivery of siRNA to tumor cells [104]. In addition, nanoparticles have the potential ability to penetrate and accumulate within tumor cells efficiently, as they have enhanced circulation time particularly when they possess hydrophilic coating on their surface, thus allowing for better therapeutics efficacy and, at the same time, minimizing the side effects of drugs. Nanoparticles can easily be imaged to track their progress in vivo. 

## 6. Current Targets for Nanoparticles-Facilitated siRNA Silencing

A combination of activated oncogenes and dysfunctional tumor suppressor genes lead to uncontrolled cell growth and blockage of natural apoptotic processes. Since crucial gene mutations responsible for cancer initiation and progression have been identified, the siRNA technology emerges as one of the highly promising approaches in treating breast cancer [105]. siRNA could be effective in cancer treatment as it is able to specifically inhibit any of the cancer-associated genes without being specific to their protein products. siRNA allows us to conceivably target the resistant cells in cancer treatment. Intriguingly, various sets of therapeutic siRNA molecules can be developed to target genes that are correlated with the multiple signaling pathways aberrantly activated in tumors. Nanotechnology is currently being explored in the development of nano-size drugs to efficiently deliver chemotherapy drugs to breast cancer cells and address the toxicity concern in relation to administration of higher doses of the drugs [106]. Introduction of exogenous siRNA into the breast cancer cells might be harnessed in order to overcome dose limitation of the chemotherapeutic drugs in clinical settings. However, since siRNA-loaded nanoparticles are accumulated in several other organs, in addition to the tumor, and the target genes that are overexpressed in breast cancer can also be expressed in those organs although at much lower level, silencing of those genes in non-target genes could produce adverse effects. Therefore, selection of target genes could be of crucial importance for clinical implications of siRNA-based nanotherapeutics. Table 1 showed the targeted genes for siRNA knockdown in breast cancer cells.

siRNA silencing of the MAPK pathway genes by targeting either the raf-1, mekk1, or mlk3 in acute myeloid leukemia (AML) cells was found to knockdown expression levels of between 40% and 60%. The data showed that when MAPK signaling pathway was partially blocked, the apoptosis pathway was upregulated and led to programmed cell death. Multiple siRNAs could be used together to target multiple genes of the MAPK pathway [107]. Bakhtiar et al. (2017) has developed nanoparticles from barium salts targeting the MAPK transcript in MCF-7 cell line and breast tumor in their in vivo study [108]. They have found success in delivery of the siRNAs, thus reducing the cell viability and inhibiting tumor growth. Silencing the MAPK genes specifically in MCF-7 cells caused the suppression of expression MAPK and activation of AKT, two important signaling molecules in both MAPK and PI3K pathways. On the other hand, our data have shown that delivery of selective siRNAs via carbonate apatite nanoparticle against the mRNA transcripts of the growth receptors including Estrogen Receptor 1 (ESR1) along with anti-apoptotic genes (BCL2), or with ERBB2 and EGFR, critically contributes to the induction of cell deaths in human and murine breast cancer cell lines by inhibiting the activation of MAPK and PI3K pathways [82]. Moreover, intravenous delivery of the selected siRNAs was able to retard tumor growth in mice.

Expressions of the BCL-2 and BCL-xL genes could promote cell survival by inhibiting apoptosis. Silencing of BCL-2 with the use of an antisense oligonucleotide appeared to be a promising cancer therapeutic approach. Silencing of Bcl-2 by siRNA followed by treatment with etoposide or doxorubicin had reduced the number of viable cancer cells and sensitized them to drug-induced apoptosis [109]. In addition, a calcium phosphate with polyelthyleneglycol (PEG)-polyanion polymer has recently been used to promote the delivery of siRNAs targeting the anti-apoptotic genes BCL-2 and BCL-xL in human breast cancer cells (MCF-7). The expression of BCL-2 and BCL-xL genes were decreased to 49% and 23%, respectively, after 48 hours of incubation with the respective siRNA. Silencing antiapoptotic genes such as BCL-2 and BCL-xL via the application of siRNAs delivered by hybrid nanoparticles was shown to be an effective and promising strategy against breast cancer [109]. 

Many cancer cells upregulate the expression of VEGF, thereby promoting angiogenesis that plays a crucial role in tumor development and metastasis. Previous study has shown that silencing VEGF expression by siRNA via polycation liposome-encapsulated calcium phosphate nanoparticles (PLCP) caused significant inhibition of tumor growth and angiogenesis in MCF-7 xenografts mice. Thus, the delivery of VEGF siRNA via PLCP to inhibit angiogenesis could be a promising strategy for breast cancer treatment, particularly when combined with DOX [110]. 

Since EpCAM, a cell surface molecule, is found to be overexpressed in cancers, Subramaniam et al. (2015) employed a novel aptamer-polyethyleneimine (PEI)-siRNA nanocomplex to target EpCAM in MCF-7 and retinoblastoma cell line (WERI-Rb1). They observed downregulation of EpCAM and inhibition of cell proliferation in the two cell lines [111]. 

Polo-like kinase 1 (PLK1) is a gene that is vital in cell division and DNA damage response and was found to be expressed in actively dividing cancer cells. PLK1 was targeted in metastatic breast cancer and triple negative breast cancer (TNBC). siRNA against the targeted gene was delivered via polymer-coated mesoporous silica with polyethyleneimine (PEI) and it was shown to inhibit cancer cell migration and invasion in TNBC cells. In the in vivo study, 80% of the target gene knockdown was observed in the mouse lung besides reduction in tumor incidence [112].

Moirangthem et al. (2016) [113] successfully transfected MDA-MB-231 human breast cancer cells with siRNA against uPA and matrix metalloproteinase 9 (MMP9) transcripts via lipid transfection. They showed that the cells were arrested in the S and G2-M after flow cytometry analysis [113]. Li et al. (2014) found that transfection of the siRNA with Lipofectamine 2000 against the cyclin-dependent kinase (CDK) 8 transcript was able to cause a significant decline in cell proliferation in MDA-MB-231 and MCF-7 cell lines [114]. Receptor tyrosine kinase expression, originated from the ROS1 oncogene, was found to be expressed in breast ductal carcinoma. Chua et al. (2013) employed carbonate apatite as the siRNA carrier targeting the c-ROS1 gene in MCF-7 cells. They observed enhancement in chemo-sensitivity of the cells towards cisplatin and paclitaxel treatments [115]. When the siRNA against IKKε was transfected into SK-BR-3 and MCF-7 human breast cancer cell lines, the siRNA was able to significantly reduce cell migration, invasion, and proliferation of both cell lines [116]. 

Silencing of the cyclin E expression by oligofectamine-facilitated siRNA delivery successfully reduced the expression of the targeted protein and led to apoptosis in SK-BR3, MDA-MB-436, and MDA-MB-157 cell lines. The xenografts of MDA-MB-436 implanted into the nude mice were successfully suppressed by cyclin E knockdown by employing the siRNA silencing technique [117].

HER2 siRNA-based therapeutics delivered using functionalized mesoporous silica nanoparticles coated with a cationic polymer and PEG conjugated to trastuzumab for HER2 targeting was shown to have an excellent safety profile. It could overcome intrinsic and acquired resistance to trastuzumab and lapatinib in HER2-positive breast cancer in vitro and in vivo [118,119]. For the long-term treatment effect of the therapeutic HER2 siRNA, the treated cells grew much slower and showed 67% increase in doubling time than cells that did receive any treatment [120]. The data indicated that the HER2 siRNA-based therapeutic provided a more durable inhibition of HER2 signaling to the cells.

## 7. Clinical Trials of nano-siRNA for Cancer Therapy

The first clinical trial of siRNA therapeutics was developed in 2004, not long after the discovery of RNAi. The rapid growth of siRNA accelerating into clinical trials is possibly due to the experience acquired during development of antisense and other nucleic acid-based therapies. To date, about 30 siRNA candidates have reached numerous stages of clinical trials for the treatment of different diseases including cancer [121,122,123]. Approximately one third of the siRNA-based therapeutics in clinical trials are targeted at cancer. 

A cyclodextrin polymer-based system designated as CALAA-01 was the first systemic siRNA delivered using targeted nanoparticles in human. The siRNA-nanoparticle formulation contained four components, which includes a duplex synthetic non-chemically modified siRNA, a cyclodextrin-containing polymer, stabilizing agent, and targeting agent that contained the human transferrin protein. The human transferrin functions as targeting ligand. CALAA-01 contained the anti-R2 siRNA targeting ribonucleotide reductase subunit M2 (RRM2) for the treatment of solid tumor. Since RRM2 regulates BCL-2 in various types of cancers and plays an active role in tumor progression, it can serve as a potential target for cancer therapy [122]. A phase-I study involving systemic administration of CALAA-01 showed that the cancer-associated gene was silenced by RNAi mechanism in target tumor cells. In addition, the patients also showed inhibition of tumor growth, as siRNA treatment was able to reduce the expression of the M2 subunit of RRM2 [122,124]. CALAA-01 treatment was carried on until the disease progressed or the treatment may no longer tolerated by the patient. About 21% of the patients discontinued the test for the reason of adverse effects. No objective tumor responses were noticed with the exception of one patient who had stable disease following four months of treatment at the dose 30 mg/m^2^ [125]. 

Besides CALAA-01, Silence Therapeutics conducted a phase-I study of the siRNA therapeutic Atu027 for the treatment of advanced solid tumor [125,126,127]. Atu027 was formulated as a liposomal particle containing siRNA that targeted protein kinase N3 (PKN3) gene. PKN3 is a downstream effector of PI3K signaling that is believed to be involved in cancer progression through the metastasis process. Inhibition of PKN3 caused significant reduction of tumor growth, as well as reduction of lymph node metastasis in vivo [128]. Early results showed that after eight weeks of treatment, Atu027 was safe in patients with advanced solid tumor with no further progression of tumors in 41% of patients [127]. Further, a phase-Ib/IIa study for Atu027 in combination with gemcitabine was achieved after the lead-in safety period [129].

ALN-VSP02 was the first dual target of a siRNA drug carried by lipid nanoparticles developed by Alnylam Pharmaceuticals (Cambridge, MA, USA). This Stable Nucleic Acid Lipid Particle (SNALP)-formulated siRNA suppressed not only VEGF, but also the cell-cycle protein kinesin spindle protein (KSP) that promoted cell-cycle arrest and, subsequently, cell death [129,130,131]. A phase-I dose-escalation study was proposed in 2009. Provisional data from pharmacodynamics measurements showed preliminary evidence of clinical efficacy in the treatment of advanced solid tumor. Nevertheless, the study did not achieve the highest tolerated dose and the trial is still ongoing, to enroll more patients in a dose-escalating manner [121]. Analysis of phase-1 clinical trials showed comparable maximum concentration and area under the curve for VEGF and KSP upon single systemic injection of ALN-VSP. The treatment normalized the tumor vasculature as determined by VEGF mRNA levels and was associated with a decrease in tumor blood flow as observed via DCE-MRI. Further, the mRNA levels of KSP that influence the mitotic cell cycle was also observed by extrahepatic tumor biopsy [125,131]. The pharmacodynamic effect observed in biopsy sample from patients validated the successful delivery of the two siRNAs, indicating stability of the nanoparticles during systemic circulation. 

A phase-I clinical trial of siRNA-EphA2-DOPC was recently authorized by the FDA and initiated by the MD Anderson Cancer Center. DOPC (1,2-dioleoylsn-glycero-3-phosphatidylcholine) is based on a type of neutral lipid to enhance siRNA entrapment efficiency. siRNA-EphA2-DOPC was constructed to shut down the activity of a genetic biomarker called EphA2. EphA2 overexpression is common in many human cancers, including breast cancer. EphA2 siRNA incorporated in DOPC nanoliposomes was greatly effective in lowering EphA2 protein levels after a single dose, and significantly reduced tumor growth three weeks after treatment [130].

Another ongoing phase-1 clinical study utilizing siRNA-transfected peripheral blood mononuclear cells (PBMCs) was APN401 for treatment of solid tumors that spread to other areas in the body or have relapsed [130]. APN401 might stop the growth of tumor cells by blocking some of the enzymes needed for cell growth. APN401 is a suspension of autologous PBMCs transfected with a siRNA that knock down Casitas-B-lineage lymphoma protein-b (Cbl-b). A single intravenous infusion of APN401 into patients with resistant solid tumors is possible and safe. This result supported phase-II clinical trials of multiple infusions of APN401 [132]. 

Other nanomedicines against breast cancer that have been approved or undergoing clinical trials were Myocet, LEP-ETU, EndoTAG-1, Lipoplatin, Genexol-PM, and Narekt-102 [133]. So far, clinical trials have shown great outcomes with no indication of adverse side effects. Moreover, avenues utilizing collateral treatment had produced promising results, hence indicating the possibility of personalized cancer treatment in the future. 

## 8. Conclusions

Chemotherapy as cancer treatment stimulates various side effects that are unbearable for the patients. RNAi technique via siRNA gene silencing should be explored in depth to further develop and enhance tumor targeting treatments, thus making it as an efficient method to combat breast cancer and other cancers. The various oncogenic genes involved in the signaling pathways in breast cancers are suitable candidates for therapeutic targets. The flexibility of employing nanoparticles to deliver siRNAs against single or multiple oncogenic genes has made the treatment strategy highly promising. Moreover, by enabling tumor-selective delivery either through passive and/or active targeting, and subsequently, promoting efficient cellular uptake, nanoparticles could be harnessed to minimize the cost of siRNAs. In addition to that, siRNA might be the solution to increase the treatment efficiency by combating classical drugs resistance.

## Figures and Tables

**Figure 1 cancers-11-00632-f001:**
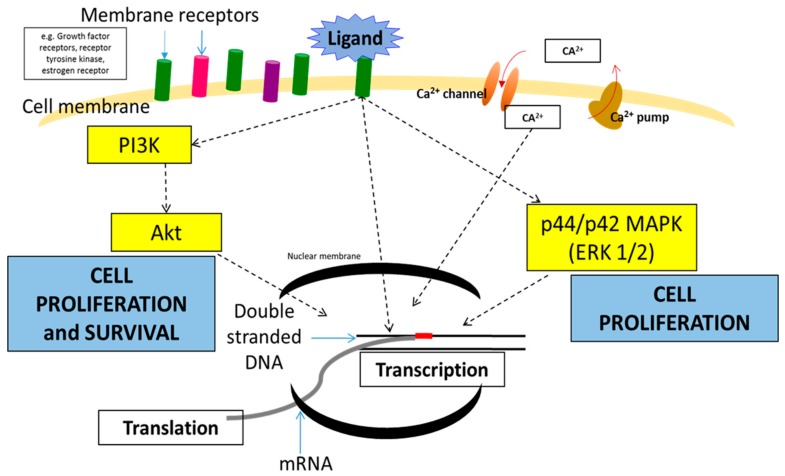
Diagram of several signaling pathways in breast cancer that lead to proliferation and survival of breast cancer cells.

**Figure 2 cancers-11-00632-f002:**
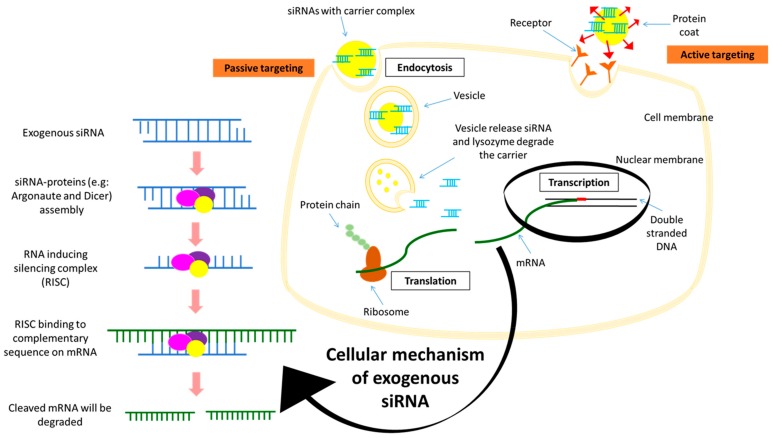
Schematic diagram of the mechanism of small interfering RNA (siRNA) in targeting mRNA for gene silencing (left) and exogenous siRNA duplex delivery into the cytoplasmic region via passive and active targeted delivery (right).

**Figure 3 cancers-11-00632-f003:**
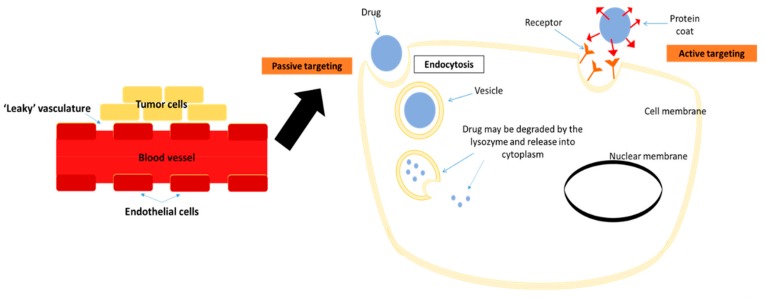
A schematic diagram showing ‘leaky’ vasculature of blood vessels at the tumor site, thus encouraging passive targeting of drug delivery. Active targeting employs the protein coating on the drug’s surface to attract receptor binding upon drug delivery into the target site.

**Table 1 cancers-11-00632-t001:** List of targeted genes for siRNA knockdown in breast cancer.

Targeted Genes	Delivery Carrier	Cell Line	Animal Model	References
***ER, BCL-2, ERBB2,* and *EGFR***	Carbonate apatite	MCF-7, MDA-MB-231	Balb/c	[82]
***egfr1* and *erbb2***	Carbonate apatite	MCF-7	Balb/c	[114]
***BCL-2* and *BCL-XL***	Calcium phosphate pEG-polyanion	MCF-7	NA	[109]
***MAPK***	Barium salts nanoparticles	MCF-7	Balb/c	[108]
***VEGF***	Polycation liposome-encapsulated calcium phosphate nanoparticles (PLCP)	MCF-7	Balb/c	[110]
***EpCAM***	Polyethyleneimine	MCF-7 and WERI-Rb1	NA	[111]
***PLK1***	Mesoporous silica coated with PEI and PEG	Bt549 and MDA-MB-231	SCID hairless SHO (Crl:SHO-Prkdc ^scid^ Hr ^hr^)	[112]
***uPA* and *MMP9***	Interferin transfection reagent	MDA-MB-231	NA	[113]
***CDK8***	Lipofectamine 2000	MDA-MB-231 and MCF-7	NA	[114]
***c-ROS1***	Carbonate apatite	MCF-7	NA	[115]
***IKKε***	Lipofectamine 2000	SK-BR-3 and MCF-7	NA	[116]
***Cyclin E***	Oligofectamine	SK-BR3, MDA-MB-157, MDA-MB-436, T47D, and MDA-MB-453	Nude mice	[117]
***HER2***	Mesoporous silica coated with cationic polymer and PEG	BT474	NA	[118]

NA: No information available.

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
