# Peer review of "Oncogenic Signaling in Tumorigenesis and Applications of siRNA Nanotherapeutics in Breast Cancer"

_cancers, 2019, doi:10.3390/cancers11050632_

Round 1

Reviewer 1 Report

It is a well written article.  However, there are some limitations in presenting the facts and need to be addressed before giving further consideration to the manuscript.  The article presents various signaling pathways for the cancer progression but fails to acknowledge the newly emerging areas such as role of glycans. The other concerns are as follows:

The authors have mentioned worldwide diagnosis of new breast cancer cases annually.  They have also commented on the expected number of new breast cancer cases in the United States.   However, they have failed to discuss the breast cancer incidence in their own country, i.e., Malaysia. 

While discussing various oncogenic pathways for cancer progression the authors have mentioned calcium signaling and claimed how the disturbance of Ca2+ homeostasis could alter the cell cycle progression.  Unfortunately, they have not identified the source of the Ca2+ and the mechanism that triggers its release. 

It seems the authors have a premeditated focus on the application of siRNA nanotherapeutics, they did not consider therapies such as Glycotherapy or the Complementary and Alternative Medicine for breast cancer treatment.  It has not been discussed adequately how the relatively short half-life of siRNAs will be over come to maintain its therapeutic window. Also, their toxicity when conjugate to metal ions.

One of the most important aspects of the review article is to provide a future direction. There are some mentioning in the conclusion but more is needed.  In addition, the cost of the therapies and the role of the pharmaceutical industries need to be considered.                 

Author Response

Reviewer 1

1)      It is a well written article.  However, there are some limitations in presenting the facts and need to be addressed before giving further consideration to the manuscript.  The article presents various signaling pathways for the cancer progression but fails to acknowledge the newly emerging areas such as role of glycans. 

Response: Thank you for your comments. We agreed on the importance of glycans as one of factors that might promotes breast cancer development. Therefore, we have added a section (3.5) of human epidermal growth factor receptor-2 (HER2), that has been associated with glycans in breast cancer progression, mainly metastasis. Moreover, we have added a study by Subramaniam et al. (2015), in section 6, on siRNA targeting the epithelial cell adhesion molecule (EpCAM) in MCF-7 and WERI-Rb1 cells.

The other concerns are as follows:

2)      The authors have mentioned worldwide diagnosis of new breast cancer cases annually.  They have also commented on the expected number of new breast cancer cases in the United States.   However, they have failed to discuss the breast cancer incidence in their own country, i.e., Malaysia. 

Response: The statistic of risk of having breast cancer of females in Malaysia has been added (Line 30).

Based on the Malaysian Nasional Cancer Registry Report (2007-2011), 1 in 30 females is having a risk of breast cancer in a lifetime”.

And the following reference has been added: Ministry of Health, National Cancer Registry Department, National Cancer Institute. (2016). Malaysian National Cancer Registry Report (2007-2011). Putrajaya: Ministry of Health Malaysia

3)      While discussing various oncogenic pathways for cancer progression the authors have mentioned calcium signaling and claimed how the disturbance of Ca2+ homeostasis could alter the cell cycle progression.  Unfortunately, they have not identified the source of the Ca2+ and the mechanism that triggers its release. 

Response: We have now added the relevant information in section 2.3.

4)      It seems the authors have a premeditated focus on the application of siRNA nanotherapeutics, they did not consider therapies such as Glycotherapy or the Complementary and Alternative Medicine for breast cancer treatment.  It has not been discussed adequately how the relatively short half-life of siRNAs will be over come to maintain its therapeutic window. Also, their toxicity when conjugate to metal ions.

Response: Since we have discussed the different signalling pathways and their keys players in initiation and progression of breast cancer, we think that RNAi technology would be the most appropriate to silence the oncogenes as part of therapeutic intervention. We have discussed the shortcomings of siRNA and strategy to overcome those limitation in section 4.2.

5)      One of the most important aspects of the review article is to provide a future direction. There are some mentioning in the conclusion but more is needed.  In addition, the cost of the therapies and the role of the pharmaceutical industries need to be considered.

Response: We have improved the conclusion accordingly.

Reviewer 2 Report

This review manuscript give us a summary on the oncogenic  signaling pathways and receptors in breast cancer. In addition, it also reviews the development of nanoparticles for siRNA delivery in pre-clinical and clinical trails. The manuscript is well organized and flows well. However, this manuscript missed a few important points which should be addressed in the revision if permitted.

1.                   In section 6-Current targets for nanoparticles-facilitated siRNA silencing, there is no analysis or insight on the current preclinical nanoparticle-based siRNA silencing research. The authors just described the key findings of the current research papers, but did not compare the advantages and limitations of different targets or different siRNA carriers.

2.                   Author’s perspective on siRNA therapeutics in breast cancer treatment is required. It is essentially important to give a prediction on the trend of this field.

Author Response

Reviewer 2

This review manuscript give us a summary on the oncogenic signaling pathways and receptors in breast cancer. In addition, it also reviews the development of nanoparticles for siRNA delivery in pre-clinical and clinical trails. The manuscript is well organized and flows well. However, this manuscript missed a few important points which should be addressed in the revision if permitted.

1. In section 6-Current targets for nanoparticles-facilitated siRNA silencing, there is no analysis or insight on the current preclinical nanoparticle-based siRNA silencing research. The authors just described the key findings of the current research papers but did not compare the advantages and limitations of different targets or different siRNA carriers.

Response:

The advantages and limitations of siRNA have been mentioned in Section 4.1 and 4.2. Section 5 discussed on the delivery system, suggesting the nanoparticles technology. However, the available literature does not provide any comparison between different targets or different nanocarriers. There are so many variables in experimental procedures, which makes it quite difficult to compare the results of two independent studies.

2. Author’s perspective on siRNA therapeutics in breast cancer treatment is required. It is essentially important to give a prediction on the trend of this field.

Response: We have edited the conclusion to address the issue. We think that it is too early to give a prediction on the trend of this field. For this it might be necessary to analyse the results of a number of completed clinical trials.

Reviewer 3 Report

Nur Izyani Kamaruzman et al describe potential molecular targets in breast cancer together with the possibility to down regulate their expression by siRNAs. Whereas the manuscript is of potential interest, the authors should carefully address some issues.

1) Section 4.2 is somehow too limited, as the authors did not describe the problem of siRNA-nanoparticles extravasation, extracellular matrix crossing as well as kidney and liver sequestration. These aspects should be at least mentioned. Moreover, the entire section should be more focused on the delivery to breast cancer.  

2) Section 5 is a general description of delivery systems for siRNA without any specific reference to the breast: the section should be made more breast-cancer-oriented.

3) In section 6 it is almost never clear if the examples mentioned consider a targeted delivery to breast cancer cells or not. Moreover, no comments about the effects of siRNA-nanoparticles on normal cells are  present. This is relevant to try to predict the general toxicity of the delivery system. In this regard, it should be reminded that available pharmacological therapies for breast cancer are administered systemically.

4) In section 7, it is necessary to explain the rational for the use of human transferrin protein in the siRNA delivery system (CALAA-01).

Author Response

Reviewer 3

Nur Izyani Kamaruzman et al describe potential molecular targets in breast cancer together with the possibility to down regulate their expression by siRNAs. Whereas the manuscript is of potential interest, the authors should carefully address some issues.

1) Section 4.2 is somehow too limited, as the authors did not describe the problem of siRNA-nanoparticles extravasation, extracellular matrix crossing as well as kidney and liver sequestration. These aspects should be at least mentioned. Moreover, the entire section should be more focused on the delivery to breast cancer.

Response: This section (4.2) is to explain the hurdles of delivery of naked siRNA. The main hurdle is the risk of degradation by the enzymes that are available in the circulation system. In section, In section 5, we discussed the roles of nanoparticles in preventing siRNA degradation and facilitating extravasation of siRNA byharnessing EPR effect. We also mentioned how hydrophilic coating of nanoparticle surface enables penetration of the nanoparticle-siRNA complex across the tumor tissue.

2) Section 5 is a general description of delivery systems for siRNA without any specific reference to the breast: the section should be made more breast-cancer-oriented.

Response: Section 5 is to establish the information regarding the siRNA delivery using nanoparticles, before we bring the readers’ attention towards siRNA-breast cancer in section 6.

3) In section 6 it is almost never clear if the examples mentioned consider a targeted delivery to breast cancer cells or not. Moreover, no comments about the effects of siRNA-nanoparticles on normal cells are present. This is relevant to try to predict the general toxicity of the delivery system. In this regard, it should be reminded that available pharmacological therapies for breast cancer are administered systemically.

Response: We have now discussed on the concerns in section 6. Regarding the toxicity of the nanoparticles in breast cancer cells, we have included new references (72, 98, 104, 106).

4) In section 7, it is necessary to explain the rational for the use of human transferrin protein in the siRNA delivery system (CALAA-01).

Response: We have added the function of human transferrin (as targeting ligand) in section 7 (in red).

Round 2

Reviewer 1 Report

A good review is the summation of existing facts, finding the gaps and provide a future direction.  The authors have led the foundation and have identified the intracellular pathways expected to be integrated with tumorigenesis.  Their summarization has scratched the surface but has failed to provide the link to tumorigenesis.  This question was raised in the prior critique for Ca2+ homeostasis.  The authors’ response is not adequate.  For example, there are more than one Ca2+ storage site in the cell and the authors did not explain which storage site is connected with tumorigenesis.  Ca2+ homeostasis is also responsible for the proliferation of normal cell.  What does it to different for tumorigenesis?  Second, Ca2+ is also associate with programed cell death, i.e., apoptosis.  It is not clear how the balance is shifted during tumorigenesis?

There are other emerging approaches to inhibit tumor growth progression.  These include glycotherapy, immunotherapy, complementary and alternative medicine, etc.  Why the authors did not consider any of these before jumping into siRNA nanotherapeutics?  When the question was raised earlier, the authors addressed by saying that co-translational modification of the N-linked glycan reduces the binding efficiency of Herceptin to HER2 and consequently reduces breast cancer progression.  This is good but does not explain the approach of glycotherapy.     

The siRNA nanotherapeutics for treating breast cancer need further explanation.  It is not clear how the authors plan to deliver the siRNA nanotherapeutics to the cancer cells/tissue.  SiRNA Without encapsulation the siRNAs will be eliminated rapidly.  If they are to be encapsulated what type of encapsulation the authors have in mind.  The liposomal formulation will increase the particle size beyond 100nm and may activate the immune response for their clearance.  Second, without targeting to tumor cells the liposomal formulation will be distrusted in other cells in the human body.  Third, the tumors are not well perfused so the claim that more nanoparticles will be located in cancer cells is misleading.

Author Response

Reviewer 1

A)    A good review is the summation of existing facts, finding the gaps and provide a future direction.  The authors have led the foundation and have identified the intracellular pathways expected to be integrated with tumorigenesis.  Their summarization has scratched the surface but has failed to provide the link to tumorigenesis.  This question was raised in the prior critique for Ca2+ homeostasis.  The authors’ response is not adequate.  For example, there are more than one Ca2+ storage site in the cell and the authors did not explain which storage site is connected with tumorigenesis.  Ca2+ homeostasis is also responsible for the proliferation of normal cell.  What does it to different for tumorigenesis?  Second, Ca2+ is also associate with programed cell death, i.e., apoptosis.  It is not clear how the balance is shifted during tumorigenesis?

Response (Round 2):

1) Thank you for your feedback. We have previously added information on the available Ca2+ storage in the cells (early of section 2.3, in red), and our major concern is overexpression of certain proteins may induce over-flow of Ca2+ and promote cancer development that differ from normal cells. We have now mentioned the consequences that having excess of Ca2+ might initiate at least one of the hallmarks of cancers as defined by Hanahan and Weinberg (2000). We have also rephrased information on a referred study (Di et al., 2015) (in red), that showed that overexpression of Rap2B protein has increased level of intracellular calcium of the breast cancer cells and induced cell proliferation, migration and invasion.

2) Endoplasmic reticulum release of Ca2+ and subsequent uptake by mitochondria involves programmed cell death. Nevertheless, an increase in Ca2+ influx activates the survival signalling pathways of the cancer cells. Cancer cells develop anti-oxidant system against the reactive oxygen species (ROS) such as hydrogen peroxide (H2O2) to maintain cells’ activities. H2O2 produced by mitochondria  mediates cysteine oxidation on transient receptor potential ankyrin 1 (TRPA1). TRPA1, a cation channel on the cell membrane, enables the up-regulation of Ca2+ into the cellular region and activates the anti-apoptotic pathway such as the PI3K/AKT signalling pathway (Reczek, C. and Chandel, N. (2018). ROS promotes cancer cell survival through calcium signaling. Cancer Cell, 33(6), pp.949-951). Data analysis from the Cancer Genome Atlas found the overexpression of TRPA1 in breast cancer.

This paragraph has been added into section 2.3 (last paragraph, line 177, in red)

B)    There are other emerging approaches to inhibit tumor growth progression.  These include glycotherapy, immunotherapy, complementary and alternative medicine, etc.  Why the authors did not consider any of these before jumping into siRNA nanotherapeutics?  When the question was raised earlier, the authors addressed by saying that co-translational modification of the N-linked glycan reduces the binding efficiency of Herceptin to HER2 and consequently reduces breast cancer progression.  This is good but does not explain the approach of glycotherapy.     

Response: Thank you for your feedback. We have added information on glycotherapy, immunotherapy, complementary and alternative medicines in the ‘Introduction’ section (in red).

C)    The siRNA nanotherapeutics for treating breast cancer need further explanation.  It is not clear how the authors plan to deliver the siRNA nanotherapeutics to the cancer cells/tissue.  SiRNA Without encapsulation the siRNAs will be eliminated rapidly.  If they are to be encapsulated what type of encapsulation the authors have in mind.  The liposomal formulation will increase the particle size beyond 100nm and may activate the immune response for their clearance.  Second, without targeting to tumor cells the liposomal formulation will be distrusted in other cells in the human body.  Third, the tumors are not well perfused so the claim that more nanoparticles will be located in cancer cells is misleading.

Response: 1) We have mentioned and listed several available vectors that have successfully delivered siRNA into cells and animal model and we have suggested that the cells will receive the encapsulated siRNA via endocytosis (Figure 2). There are also studies that we have cited, reporting intravenous injection of the nanoparticles-siRNA in animal models.

Regarding the size of the vector-siRNA complex, we have mentioned the importance of nano-technology to create nano-sized vector, that can successfully deliver the siRNA without triggering the immune response and causing toxicity effects.

2) We have addressed this in line 361 (in red)

3) As discussed in section 5 and illustrated in Fig. 3, due to enhanced permeability and retention (EPR) effect, nanoparticles are accumulated in the tumor tissue which has leaky vasculature and underdeveloped lymphatic system.

Reviewer 3 Report

1) Section 4.2 is somehow too limited, as the authors did not describe the problem of siRNA-nanoparticles extravasation, extracellular matrix crossing as well as kidney and liver sequestration. These aspects should be at least mentioned. Moreover, the entire section should be more focused on the delivery to breast cancer.

Response: This section (4.2) is to explain the hurdles of delivery of naked siRNA. The main hurdle is the risk of degradation by the enzymes that are available in the circulation system. In section, In section 5, we discussed the roles of nanoparticles in preventing siRNA degradation and facilitating extravasation of siRNA by harnessing EPR effect. We also mentioned how hydrophilic coating of nanoparticle surface enables penetration of the nanoparticle-siRNA complex across the tumor tissue.

I still think this section could be improved by at least mentioning the problem of  extravasation, extracellular matrix crossing as well as kidney and liver sequestration

2) Section 5 is a general description of delivery systems for siRNA without any specific reference to the breast: the section should be made more breast-cancer-oriented.

Response: Section 5 is to establish the information regarding the siRNA delivery using nanoparticles, before we bring the readers’ attention towards siRNA-breast cancer in section 6.

Also in this case it could be enough mentioning some breast-specify delivery  systems.

3) In section 6 it is almost never clear if the examples mentioned consider a targeted delivery to breast cancer cells or not. Moreover, no comments about the effects of siRNA-nanoparticles on normal cells are present. This is relevant to try to predict the general toxicity of the delivery system. In this regard, it should be reminded that available pharmacological therapies for breast cancer are administered systemically.

 Response: We have now discussed on the concerns in section 6. Regarding the toxicity of the nanoparticles in breast cancer cells, we have included new references (72, 98, 104, 106).

OK

4) In section 7, it is necessary to explain the rational for the use of human transferrin protein in the siRNA delivery system (CALAA-01).

Response: We have added the function of human transferrin (as targeting ligand) in section 7 (in red).

OK

Author Response

Reviewer 3

1) Section 4.2 is somehow too limited, as the authors did not describe the problem of siRNA-nanoparticles extravasation, extracellular matrix crossing as well as kidney and liver sequestration. These aspects should be at least mentioned. Moreover, the entire section should be more focused on the delivery to breast cancer.

Response: This section (4.2) is to explain the hurdles of delivery of naked siRNA. The main hurdle is the risk of degradation by the enzymes that are available in the circulation system. In section 5, we discussed the roles of nanoparticles in preventing siRNA degradation and facilitating extravasation of siRNA by harnessing EPR effect. We also mentioned how hydrophilic coating of nanoparticle surface enables penetration of the nanoparticle-siRNA complex across the tumor tissue.

I still think this section could be improved by at least mentioning the problem of  extravasation, extracellular matrix crossing as well as kidney and liver sequestration

Response: We have improved the section by adding roles of kidneys and endosomal fusion with lysosomes in elimination of naked siRNAs. Moreover, it has been mentioned that since nanoparticles are prone to interact with reticuloendothelial system (RES), they require surface modification prior to being used for systemic delivery of siRNA

 Response (Round 2):

2) Section 5 is a general description of delivery systems for siRNA without any specific reference to the breast: the section should be made more breast-cancer-oriented.

Response: Section 5 is to establish the information regarding the siRNA delivery using nanoparticles, before we bring the readers’ attention towards siRNA-breast cancer in section 6.

Also in this case it could be enough mentioning some breast-specify delivery systems.

Response: We have mentioned in the section that ‘it is crucial to identify highly expressed receptors, particularly on the breast cancer cells to increase their specific binding with the ligands on the siRNA complexes and successful delivery of siRNA to the targeted sites’.

 Response (Round 2):

3) In section 6 it is almost never clear if the examples mentioned consider a targeted delivery to breast cancer cells or not. Moreover, no comments about the effects of siRNA-nanoparticles on normal cells are present. This is relevant to try to predict the general toxicity of the delivery system. In this regard, it should be reminded that available pharmacological therapies for breast cancer are administered systemically.

 Response: We have now discussed on the concerns in section 6. Regarding the toxicity of the nanoparticles in breast cancer cells, we have included new references (72, 98, 104, 106).

OK

 Response (Round 2): Thank you

4) In section 7, it is necessary to explain the rational for the use of human transferrin protein in the siRNA delivery system (CALAA-01).

Response: We have added the function of human transferrin (as targeting ligand) in section 7 (in red).

OK

Response (Round 2): Thank you

Round 3

Reviewer 1 Report

page 1, line 32; add the mortality rate due to breast cancer in Malaysia.

page 1, line 42: replace "human epidermal receptor 2" with human epidermal growth factor receptor 2

page 11, line 395; replace the word "drugability" with an appropriate word.

The article may help designing new breast cancer therapeutics without spinning on the same axis of the currently available drugs.

Author Response

Reviewer 1, Round 3.

1)      page 1, line 32; add the mortality rate due to breast cancer in Malaysia.

Response: Thank you. We have added the mortality rate of breast cancer in Malaysia (in red).

The mortality rate of breast cancer in Malaysia is estimated to be ~16.7 to 20 in 100,000 [4]’

2)      page 1, line 42: replace "human epidermal receptor 2" with human epidermal growth factor receptor 2

Response: Thank you. Changes have been made as suggested.

3)      page 11, line 395; replace the word "drugability" with an appropriate word.

Response: Thank you. The word has been changed.

4)      The article may help designing new breast cancer therapeutics without spinning on the same axis of the currently available drugs.

Response: Thank you.